# Functional divergence of Plexin B structural motifs in distinct steps of *Drosophila* olfactory circuit assembly

Ricardo Guajardo[1], David J Luginbuhl[1], Shuo Han[2], Liqun Luo[1]*, Jiefu Li[1]

[1]Department of Biology, Howard Hughes Medical Institute, Stanford University, Stanford, United States; [2]Department of Chemistry, Stanford University, Stanford, United States

**Abstract** Plexins exhibit multitudinous, evolutionarily conserved functions in neural development. How Plexins employ their diverse structural motifs in vivo to perform distinct roles is unclear. We previously reported that Plexin B (PlexB) controls multiple steps during the assembly of the *Drosophila* olfactory circuit (Li et al., 2018b). Here, we systematically mutagenized structural motifs of PlexB and examined the function of these variants in these multiple steps: axon fasciculation, trajectory choice, and synaptic partner selection. We found that the extracellular Sema domain is essential for all three steps, the catalytic site of the intracellular RapGAP is engaged in none, and the intracellular GTPase-binding motifs are essential for trajectory choice and synaptic partner selection, but are dispensable for fasciculation. Moreover, extracellular PlexB cleavage serves as a regulatory mechanism of PlexB signaling. Thus, the divergent roles of PlexB motifs in distinct steps of neural development contribute to its functional versatility in neural circuit assembly.
DOI: https://doi.org/10.7554/eLife.48594.001

*For correspondence:
lluo@stanford.edu

Competing interests: The authors declare that no competing interests exist.

## Introduction

Nervous systems are composed of intricately structured assemblies of neurons. Indeed, their proper function requires highly specified circuit organization, wherein neurons make precise connections with their synaptic partners. The study of neural circuit assembly has generated an ever-expanding catalog of wiring molecules, whose biological roles ensure the fidelity of neuronal connections and thus of information transmission (*Hong and Luo, 2014*; *Jan and Jan, 2010*; *Kolodkin and Tessier-Lavigne, 2011*; *Li et al., 2018a*; *Sanes and Yamagata, 2009*; *Zipursky and Sanes, 2010*). While structural and biophysical studies have advanced our understanding of the atomic architectures of these wiring molecules, for most of them it remains largely unknown how their structural motifs behave in specific neurodevelopmental processes in vivo.

Plexins, a conserved family of single-pass transmembrane receptors, play varied roles in the development and homeostasis of diverse tissues in both vertebrates and invertebrates. Through the effort of many laboratories in the past two decades (*Alto and Terman, 2017*; *Koropouli and Kolodkin, 2014*; *Kruger et al., 2005*; *Pascoe et al., 2015*; *Pasterkamp, 2012*; *Siebold and Jones, 2013*; *Worzfeld and Offermanns, 2014*), genetic functions and biochemical properties of Plexins have been substantially characterized. However, even for Plexins, we have sparse knowledge on the connection between their structural motifs and their in vivo cellular functions, especially in the context of multi-step neural circuit assembly.

Over 600 million years old, Plexin-family receptors display high degrees of conservation across evolutionarily distant species for both extracellular and cytoplasmic domains (*Junqueira Alves et al., 2019*). Structural and biochemical investigations have identified several core domains required for

Plexin signaling (*Bell et al., 2011*; *He et al., 2009*; *Janssen et al., 2010*; *Janssen et al., 2012*; *Kong et al., 2016*; *Liu et al., 2010*; *Nogi et al., 2010*; *Shang et al., 2017*; *Tong et al., 2009*; *Tong et al., 2007*; *Tong et al., 2008*; *Wang et al., 2012*; *Wang et al., 2013*): the extracellular Sema domain, the intracellular Rac and Rho GTPase-binding sites, and the intracellular catalytic RapGAP domain (*Figure 1A*). Besides the Mical and CRMP (collapsin response mediator protein) pathways for certain Plexins (*Alto and Terman, 2017*), the current working model of Plexin signal transduction suggests that concomitant Semaphorin extracellular binding and Rac intracellular binding lead to Plexin dimerization, which subsequently activates the catalytic GAP domain to hydrolyze Rap-GTP for downstream signaling (*Pascoe et al., 2015*). However, the in vivo functional relevance of this model has yet to be determined. Notably, the catalytic GAP domain has been shown to be essential in neural tube closure (*Worzfeld et al., 2014*) but not in motor axon guidance (*Yang et al., 2016*), suggesting that Plexins may use structural motifs differentially in distinct developmental contexts. Moreover, certain Plexin motifs have yet to be functionally characterized. We particularly note the convertase cleavage site – a conserved signature of all class B Plexins (*Artigiani et al., 2003*). Despite its conservation across hundreds of millions of years, to our knowledge no biological function has ever been reported since its discovery over a decade ago. Considering that the cleavage event breaks the Plexin protein and thus physically separates the extracellular and transmembrane-cytoplasmic parts, it can activate, inactivate, or serve a more complex regulatory role in Plexin B signaling.

We recently reported that PlexB plays indispensable roles in multiple steps during the assembly of the *Drosophila* olfactory map, with level-independent functions in the axon fasciculation of olfactory receptor neurons (ORNs) and level-dependent tasks in ORN axon trajectory choice and subsequent glomerular targeting (*Li et al., 2018b*). These findings reveal that, within one system, PlexB regulates several fundamental cellular processes of neural wiring, namely axon-axon interaction (fasciculation), axon guidance (trajectory choice), and synaptic partner selection (glomerular targeting) (*Figure 1C*). Given that this level dependence is only observed in trajectory choice and glomerular targeting but not in fasciculation, it is likely that PlexB signals in different ways when executing these distinct tasks. The multi-step development of the fly olfactory map thus provides an excellent system for characterizing the structure-to-function relationship of Plexin in vivo.

Through systematic mutagenesis of PlexB structural motifs (*Figure 1A,B*) and functional interrogation in the context of fly olfactory circuit assembly (*Figure 1C*), we report here the differential engagement of PlexB structural motifs in distinct neurodevelopmental processes. From the global necessity of Sema domain to the overall expendability of GAP catalytic integrity, as well as the involvement of GTPase-binding motifs in trajectory choice and glomerular targeting but not in fasciculation, our findings link the categorical diversity of PlexB-dependent wiring processes to its varied utilization of distinct signaling modules. Moreover, we identified a regulatory role of PlexB cleavage in vivo and surprisingly found that the cleaved fragments can functionally reconstitute for signaling. Collectively, our analysis reveals how a single molecule, PlexB, plays multitudinous roles in instructing cellular behaviors through the varied use of its distinct structural motifs.

## Results

### Systematic mutagenesis of PlexB structural motifs and in vivo functional assays

To dissect the structure-to-function relationship of PlexB in the assembly of the fly olfactory map, we generated nine UAS transgenic lines carrying PlexB variants (described below; *Figure 1B*) produced by site-directed mutagenesis. A V5 tag was added to the C-terminus of each variant. To obtain comparable expression levels, all transgenes were mutagenized from a single wild-type *UAS-PlexB* construct (*Joo et al., 2013*) and integrated into the same genomic locus, *ZH-attP-86Fb* (*Bischof et al., 2007*). Their expression in *Drosophila* neurons in vivo was verified by Western blotting with an anti-V5 antibody (*Figure 1—figure supplement 1*).

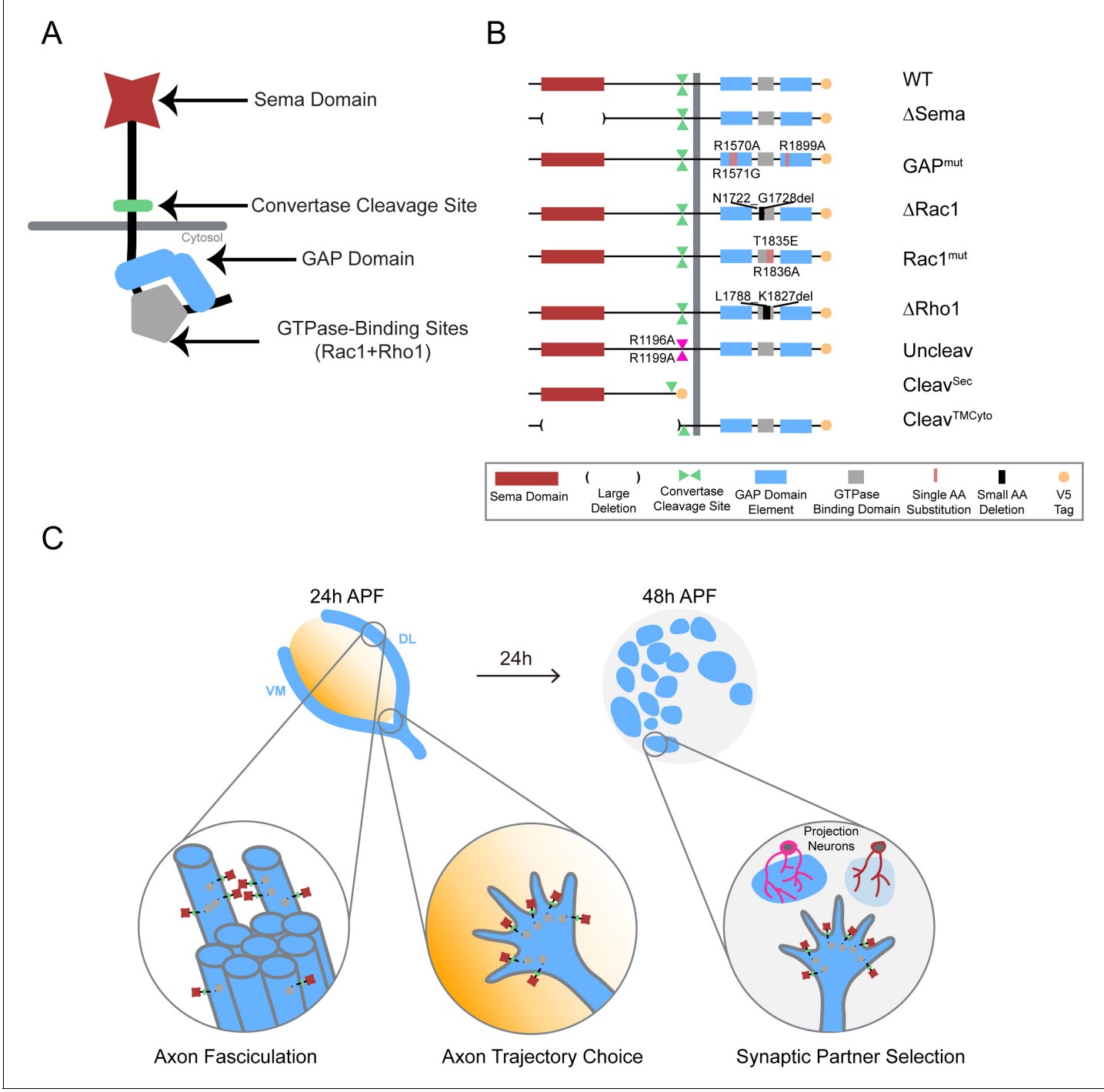

**Figure 1.** Systematic mutagenesis of PlexB structural motifs and functional interrogation in the stepwise assembly of the *Drosophila* olfactory map. (**A**) The PlexB protein consists of several conserved structural motifs, including an extracellular Sema domain, a juxtamembrane convertase cleavage site, a cytoplasmic GTPase-binding region for Rac1 and Rho1, and a cytoplasmic bipartite GAP domain. (**B**) Schematic summary of PlexB variants generated in this study. Each variant encodes either a mutated form of PlexB with one structural motif disrupted or a cleaved product of PlexB. (**C**) In the developing antennal lobe, ORN axons first fasciculate into bundles. Each ORN axon chooses a defined trajectory along the edge of the antennal lobe, in part responding to the extracellular Sema-2a/2b gradients (orange). Subsequently, ORN axons innervate the antennal lobe to interact with dendrites of prospective projection neuron partners and thus establish specific synaptic connections. PlexB participates in all these processes (*Li et al., 2018b*), providing an in vivo platform for examining the functionality of PlexB variants in multiple, distinct wiring steps.

DOI: https://doi.org/10.7554/eLife.48594.002

The following figure supplements are available for figure 1:

*Figure 1 continued on next page*

*Figure 1 continued*

**Figure supplement 1.** Western blot showing that PlexB variants are stably expressed in vivo.
DOI: https://doi.org/10.7554/eLife.48594.003
**Figure supplement 2.** Homology alignment of Plexins from different species and sub-families showing the conserved arginine residues (highlighted in red) of the GAP domain.
DOI: https://doi.org/10.7554/eLife.48594.004

## Sema Domain

The extracellular Sema domain is a molecular signature of Plexins and Semaphorins – the canonical Plexin ligands. It mediates binding between Plexins and Semaphorins and triggers Plexin signal transduction (*Janssen et al., 2010*; *Nogi et al., 2010*). PlexB with its entire Sema domain deleted (ΔSema) was expressed and cleaved normally in vivo (*Figure 1—figure supplement 1*).

## Catalytic RapGAP domain

A bipartite GTPase-activating protein (GAP) domain resides in the cytoplasmic part of Plexins. Recent studies identified the Ras homolog Rap as its substrate for signal transduction (*Wang et al., 2012*; *Wang et al., 2013*). With homology alignment (*Figure 1—figure supplement 2*), we identified the three arginine residues that are essential for the GAP catalytic activity (*Wang et al., 2012*; *Worzfeld et al., 2014*) in fly PlexB and generated a variant with all three arginines mutated (R1570A, R1571G, R1899A; noted as GAP$^{mut}$). These point mutations did not affect PlexB's expression and cleavage in vivo (*Figure 1—figure supplement 1*).

## GTPase-binding sites

Sitting between the two arms of the RapGAP domain, the GTPase-binding region of fly PlexB has been shown to interact with small GTPases Rac1 and Rho1 (*Hu et al., 2001*). Within this region, two phylogenetically conserved sites mediate the PlexB-Rac1 interaction. Structural analysis found that class B Plexins can interact simultaneously with two Rac1 molecules, with one bound at each site (*Bell et al., 2011*). Thus, we built two PlexB variants that independently disrupt the Rac1-binding sites: ΔRac1, which contains a small deletion in the first Rac1-binding site (*Hu et al., 2001*), and Rac1$^{mut}$, which has two amino acid substitutions (T1835E, R1836A) in the second site (*Bell et al., 2011*). We also generated a small deletion (ΔRho1) that abolishes the PlexB-Rho1 interaction (*Hu et al., 2001*).

Intriguingly, while full-length PlexB proteins were present normally, the cleaved C-terminal fragment was markedly reduced in ΔRac1 or ΔRho1 variants (*Figure 1—figure supplement 1*), as previously observed in cell culture (*Artigiani et al., 2003*). We will elaborate on this observation in the context of developmental function in the Discussion section.

## Cleavage site

The functionally uncharacterized cleavage is a conserved signature of all class B Plexins from flies to mammals. In developing fly brains, only a small fraction of endogenous PlexB proteins are present in the full-length form (*Li et al., 2018b*). Notably, the cleaved C-terminal fragment is not degraded in vivo. This is consistent with a previous observation that cleaved PlexB products stably associate in a complex in cell culture (*Artigiani et al., 2003*), suggesting that cleaved PlexB may be functional in signaling. To investigate the function of PlexB cleavage in vivo, we generated a PlexB variant with its cleavage sites mutated (R1196A, R1199A; noted as Uncleav), as well as constructs expressing cleaved N-terminal and C-terminal products (Cleav$^{Sec}$ and Cleav$^{TMCyto}$, respectively). Indeed, the two arginine mutations abolished PlexB cleavage and increased the presence of full-length PlexB (*Figure 1—figure supplement 1*). We also observed two faint bands around 100 kDa, the pattern of which was distinct from the original cleavage (*Figure 1—figure supplement 1*). PlexB possibly undergoes atypical processing when the convertase site is mutated. The cleaved fragments, when individually expressed, were also stable in vivo (*Figure 1—figure supplement 1*).

To determine the developmental function of these PlexB structural motifs, we tested the efficacy of these variants at recapitulating the activity of wild-type PlexB in multiple wiring tasks in developing fly olfactory receptor neurons: axon fasciculation, axon trajectory choice, and synaptic partner

selection (*Figure 1C*). While all of them are PlexB-dependent, these wiring processes occur sequentially and can be assayed independently (*Li et al., 2018b*), thus providing a platform for examining the functional engagement of individual motifs in distinct developmental tasks.

## Axon fasciculation

At about 18 hours after puparium formation (hAPF), ORN axons arrive at the antennal lobe and fasciculate with neighboring axons, forming two discrete bundles. Over the next 6 hr, these two axon bundles circumnavigate the antennal lobe (*Figure 2A*, left panel). Previously, we found that ORN axon fasciculation is mediated by PlexB-dependent axon-axon interactions (*Joo et al., 2013*; *Li et al., 2018b*). In PlexB loss-of-function mutants (*plexB*$^{-/-}$), ORN axon defasciculation was observed in almost every antennal lobe with differing severity (*Figure 2A*, middle and right panels; *Figure 2B*). To quantify the fasciculation defects, we blindly binned 24hAPF antennal lobes into one of the three following categories: 1) no defasciculation (*Figure 2A*, left panel); 2) mild defasciculation, in which fasciculation defects were present but axon bundles that normally circumnavigate the antennal lobe were clearly preserved (*Figure 2A*, middle panel); and 3) severe defasciculation, where pronounced invasion of the central antennal lobe by ORN axons was observed, along with the loss of axon bundles (*Figure 2A*, right panel). In *plexB*$^{-/-}$ flies, expression of a wild-type *PlexB* transgene in ORNs significantly restored axon fasciculation (*Figure 2B*). However, ORN-specific rescue was not complete (*Figure 2B*), suggesting that PlexB supplied by other cellular sources may also contribute to ORN axon fasciculation. Nonetheless, the rescue assay provides a quantifiable readout with a large dynamic range to examine if each PlexB structural motif participates in axon fasciculation.

As shown in *Figure 2C*, the rescue by Sema domain-deleted PlexB completely failed, indicating the necessity of the Sema domain in PlexB-mediated fasciculation. On the other hand, none of the intracellular motifs we assayed were required, as none of the mutants displayed a compromised ability to rescue fasciculation defects (*Figure 2D–G*). These data suggest that either the cytoplasmic signaling is not required for fasciculation, or that different motifs play redundant roles in mediating fasciculation.

Notably, uncleavable PlexB appeared to exhibit better rescue than wild-type PlexB, with only one severe defasciculation case out of 47 examined (*Figure 2H*). Considering that uncleavable PlexB only supplies full-length PlexB proteins, this finding suggests that the full-length PlexB proteins play a predominant role in mediating axon fasciculation. Consistently, neither the N- nor C- terminal cleaved products, when expressed separately (*Figure 2I,J*) or together (*Figure 2K*), exhibited any rescue effects.

Taken together, our data suggest that PlexB-dependent axon fasciculation is mediated by full-length PlexB but not the cleaved fragments. Moreover, fasciculation appears to not require any individual cytoplasmic signaling motif but relies on the extracellular Sema domain. Thus, PlexB-dependent axon fasciculation is likely an intercellular adhesion process, in which full-length PlexB proteins bundle axons together through Sema domain-mediated molecular adhesion.

## Axon trajectory choice

After their arrival at the ventrolateral corner of the antennal lobe at around 18hAPF, individual ORN axons choose one of the two trajectories – dorsolateral (DL) or ventromedial (VM) – and then circumnavigate the antennal lobe in the next 6 hours (*Figure 3A*; left panels) (*Jefferis et al., 2004*). Importantly, axons of each ORN class stereotypically choose one defined trajectory. We previously found that trajectory choice is regulated by the axonal PlexB level: a high PlexB level drives axons to the DL trajectory while a low PlexB level confers a VM choice (*Li et al., 2018b*). Consequently, PlexB overexpression in ORNs shifts axons to the DL trajectory (*Figure 3A*; right panels) (*Li et al., 2018b*). This PlexB level-dependent trajectory choice thus provides an opportunity to examine the involvement of individual structural motifs in an axon guidance task. Theoretically, this could also be examined in a rescue context, in which the functionality of PlexB variants is tested in a *plexB* null background. However, *plexB* null mutants exhibit axon fasciculation defects that cannot be completely rescued even by wild-type PlexB, as described above (*Figure 2B*). The abundance of defasciculated axons precludes proper quantification of trajectory choice, making it impracticable to test the structural motifs in a rescue assay. Thus, we assessed the functional engagement of PlexB

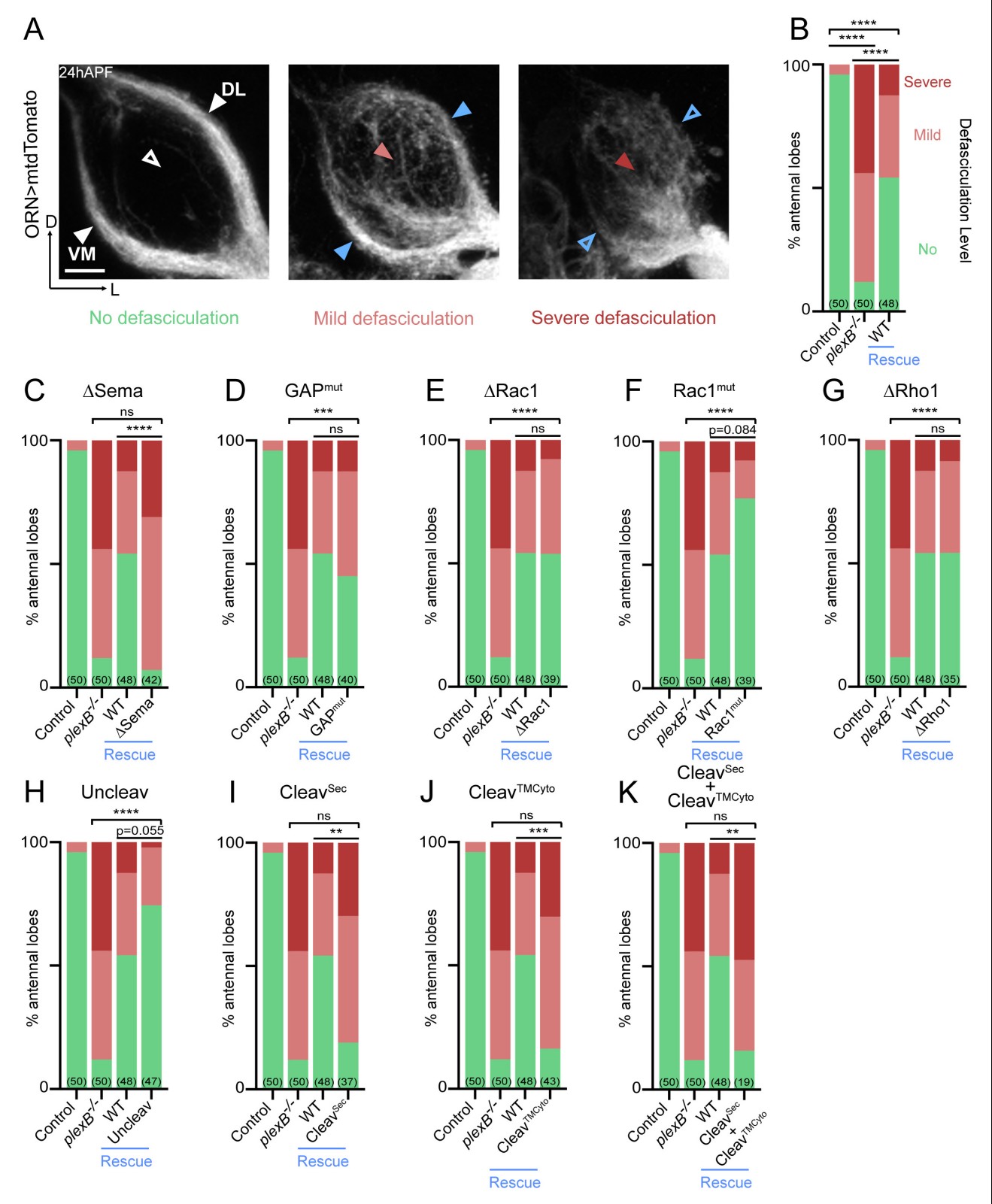

**Figure 2.** Axon fasciculation requires full-length PlexB but not its cytoplasmic motifs individually. (**A**) In a *wild-type* fly brain at 24hAPF, ORN axons fasciculate into two bundles (left panel; white arrowheads) surrounding the antennal lobe without innervating it (left panel; empty white arrowhead). Loss of PlexB (*plexB$^{-/-}$*) causes defasciculation of ORN axons with differing severity (middle and right panels; red arrowheads). In the severe cases, axon bundles are completely missing (right panel; empty blue arrowheads). ORN axons were labeled by pan-ORN *Peb-GAL4* (*Sweeney et al., 2007*) driven

*Figure 2 continued on next page*

*Figure 2 continued*

mtdTomato expression. (**B**) Quantification of fasciculation defects by binning antennal lobes into three categories – no, mild, and severe defasciculation. Expressing wild-type PlexB in ORNs significantly but not completely restores ORN axon fasciculation in *plexB* mutant flies. 'Rescue' hereafter denotes ORN-specific expression of PlexB variants in *plexB$^{-/-}$* flies. (**C–K**) Quantification of fasciculation defects in ORN-specific rescue experiments with respective PlexB variants. Sample sizes are noted in parentheses. Significance of the contingency tables in *Figure 2B–K* was determined by Fisher's exact test. ns, not significant; **p<0.01; ***p<0.001; ****p<0.0001. Images are shown as maximum z-projections of confocal stacks. Scale bars, 10 µm. Axes, D (dorsal), L (lateral).

DOI: https://doi.org/10.7554/eLife.48594.005

structural motifs in trajectory choice by examining overexpression-induced DL shifting of ORN axons.

As described previously (*Li et al., 2018b*), we quantified the trajectory choice by a ratio of the fluorescence intensity of DL and VM axons (DL/VM; *Figure 3B*). Recapitulating our previous observations, overexpression of wild-type PlexB in ORNs drove axons to the DL bundle, raising the mean DL/VM ratio to 0.94 from 0.67 of controls (*Figure 3A,B*). We note that the DL shifting observed here was not as severe as we previously reported (*Li et al., 2018b*), probably due to lower expression of the newly generated transgene, which was inserted at a genomic locus different from that of the randomly integrated transgene used previously.

Between 18–24hAPF, the canonical ligands of PlexB—Sema2a and Sema2b—establish a gradient along the VM-to-DL axis to instruct PlexB-expressing ORN axons in trajectory choice (*Joo et al., 2013*). In line with this, deletion of the Sema domain that mediates the Plexin-Semaphorin interaction completely disrupted the ability of PlexB to drive a DL shift (*Figure 3C*). Regarding the cytoplasmic motifs, mutating either Rac1- or Rho1- binding sites impaired, at least partially, the DL shift caused by PlexB overexpression (*Figure 3E–G*), revealing the functional necessity of PlexB-GTPase interactions in trajectory choice. Notably, while deleting the first of the Rac1-binding sites (ΔRac1) entirely abolished the DL shift caused by PlexB overexpression (*Figure 3E*), mutating the second Rac1-binding site (Rac1$^{mut}$) only partially weakened the phenotype (*Figure 3F*), suggesting the differential importance of these sites in mediating the PlexB-Rac1 interaction. Interestingly, the catalytic RapGAP domain was not required (*Figure 3D*), as in PlexA-dependent motor axon guidance (*Yang et al., 2016*).

We then examined the involvement of PlexB cleavage in trajectory choice and found that uncleavable PlexB was significantly more potent than wild-type PlexB at driving DL shift (*Figure 3H*), suggesting that full-length PlexB proteins are more active for this function. Considering that trajectory choice is a PlexB level-dependent process (*Li et al., 2018b*), the cleavage of PlexB thus limits the level of full-length PlexB proteins (*Figure 1—figure supplement 1*) and can potentially regulate the fidelity of trajectory choice. Intriguingly, while the cleaved fragments did not show any function individually (*Figure 3I,J*), simultaneous expression of both partially but significantly promoted DL shift of ORN axons (*Figure 3K*), indicating the functional reconstitution of cleaved fragments in vivo for signaling in trajectory choice. In line with this, a previous study observed that the cleaved Plexin B products are biochemically associated with each other as a complex (*Artigiani et al., 2003*).

In summary, distinct from its adhesion-like function in axon fasciculation (*Figure 2*), PlexB instructs axon trajectory choice by engaging both extracellular and cytoplasmic components, although the RapGAP catalytic activity appears dispensable. Cleavage of PlexB can regulate the strength of PlexB signaling, as the full-length protein appeared to be more active, while the reconstituted fragments also exhibited activity in trajectory choice.

## Synaptic partner selection

Between 24–48hAPF, ORN axons innervate the antennal lobe and search for the dendrites of their synaptic partners—olfactory projection neurons (PNs). By 48hAPF, the antennal lobe has been divided into roughly 50 proto-glomeruli, where the axons and dendrites of matching ORNs and PNs interact (*Jefferis et al., 2004*). We previously found that PlexB plays a level-dependent role in instructing glomerular selection of ORN axons, independently of trajectory choice (*Li et al., 2018b*). Specifically, ORN axons targeting to several discrete glomeruli express higher levels of PlexB than their neighbors. These PlexB-high glomeruli localize mainly in the medial antennal lobe, such as DM1 and DM5, along with a few scattered glomeruli, like VA4. PlexB overexpression in ORNs

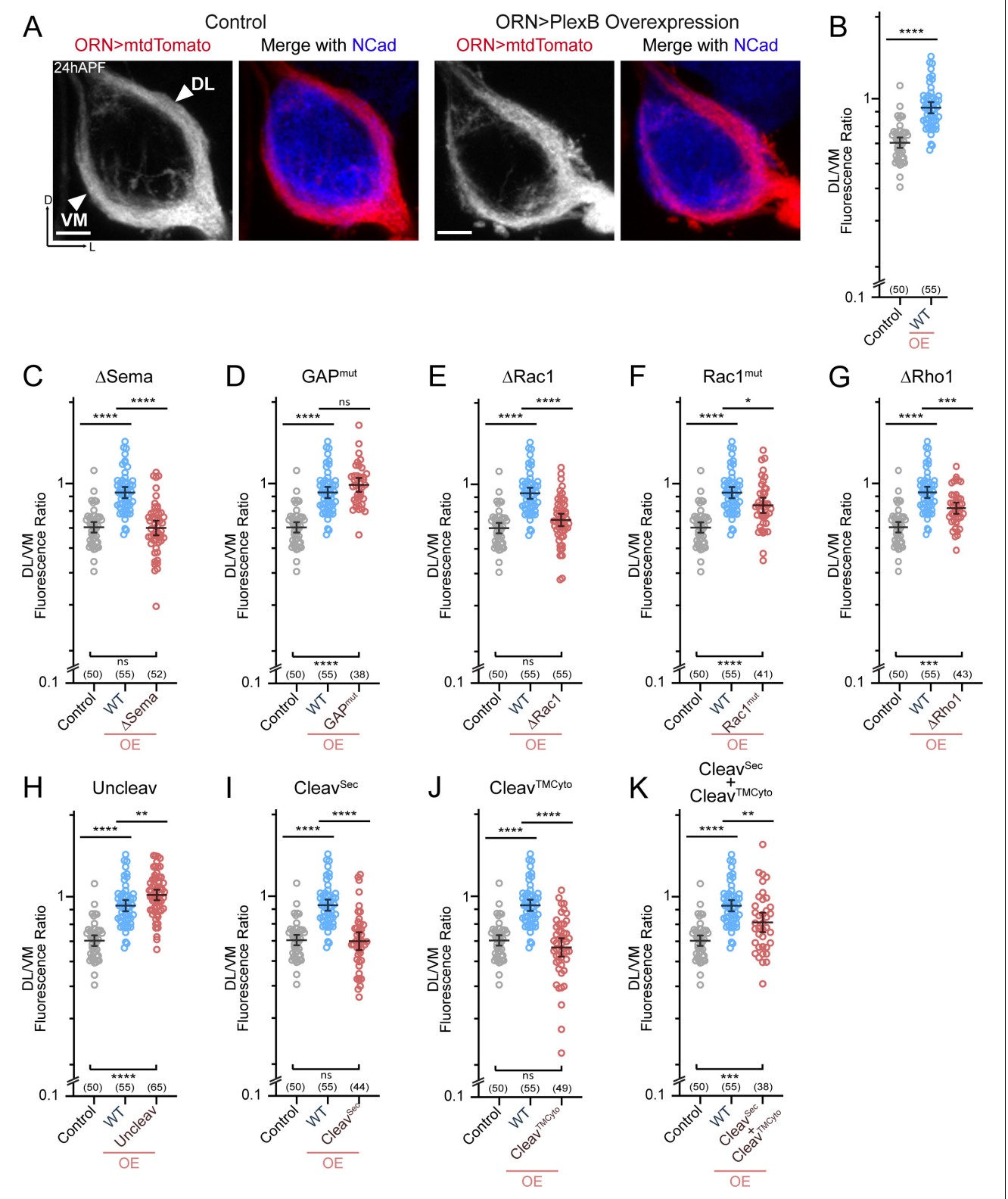

**Figure 3.** ORN trajectory choice requires both extracellular and cytoplasmic modules of PlexB. Both full-length and reconstituted fragments of PlexB transduce signal in trajectory choice. (**A**) In *wild-type* pupal brains at 24hAPF, ORN axons form the dorsolateral (DL) and ventromedial (VM) trajectories circumnavigating the antennal lobe (left panels). Overexpression of PlexB in ORNs shifts ORN axons to the DL trajectory (right panels). ORN axons were labeled by pan-ORN *Peb-GAL4* (**Sweeney et al., 2007**) driven mtdTomato expression. Antennal lobes were co-stained with a neuropil marker

*Figure 3 continued on next page*

*Figure 3 continued*

N-cadherin (NCad). (**B**) Fluorescence intensity ratios of ORN axon trajectories (DL/VM) in *wild-type* and PlexB overexpression brains at 24hAPF. Geometric means: control, 0.68; WT OE, 0.94. 'OE' hereafter denotes ORN-specific overexpression of PlexB variants. (**C–K**) Fluorescence intensity ratios of ORN axon trajectories (DL/VM) for respective PlexB variants. Geometric means: ΔSema, 0.68; GAP$^{mut}$, 1.01; ΔRac1, 0.73; Rac1$^{mut}$, 0.84; ΔRho1, 0.81; Uncleav, 1.04; Cleav$^{Sec}$, 0.69; Cleav$^{TMCyto}$, 0.65; Cleav$^{Sec}$ + Cleav$^{TMCyto}$, 0.82. Sample sizes are noted in parentheses. Significance among multiple groups in *Figure 3B–K* was determined by one-way ANOVA with Tukey's test for multiple comparisons. ns, not significant; *$p<0.05$; **$p<0.01$; ***$p<0.001$; ****$p<0.0001$. Images are shown as maximum z-projections of confocal stacks. Scale bars, 10 μm. Axes, D (dorsal), L (lateral).
DOI: https://doi.org/10.7554/eLife.48594.006

promotes mistargeting of ORN axons to PlexB-high glomeruli, while RNAi-based knockdown shows the opposite preference (*Li et al., 2018b*).

In line with our previous observations, PlexB overexpression caused mistargeting of VA2 ORN axons stereotypically to the DM5 glomerulus (*Figure 4A,B*). Like the weakened DL shift in trajectory choice due to the new UAS transgene (*Figure 3B*), we also note that the mistargeting preference changed from VA4 to DM5, whose PlexB level is lower than VA4 (*Li et al., 2018b*). Similarly, the phenotypic penetrance of mistargeting dropped to about 30% (*Figure 4B*) from the original 70% (*Li et al., 2018b*). Nonetheless, the glomerular mistargeting caused by PlexB overexpression provides a clear and quantifiable readout for examining the functional engagement of PlexB structural motifs in synaptic partner selection.

PlexB without its Sema domain failed to induce glomerular mistargeting (*Figure 4C*), emphasizing the global necessity of Sema domain in all wiring processes examined. In contrast, the RapGAP catalytic site was not essential for any tested processes, including synaptic partner selection (*Figure 4D*). As in trajectory choice, glomerular targeting required PlexB-GTPase interactions, as the variants disrupting Rac1- or Rho1- binding sites substantially reduced glomerular mistargeting events caused by PlexB overexpression (*Figure 4E–G*). Notably, deletion of the first Rac1-binding site again resulted in greater functional disruption than substitution at the second site (*Figure 4E,F*; as well as *Figure 3E,F*), further supporting the differential importance of these two regions in bridging PlexB and Rac1.

Overexpression of the cleaved products, either independently or together, was insufficient to drive mistargeting (*Figure 4I–K*), suggesting that synaptic partner selection is likely mediated by the full-length PlexB. However, uncleavable PlexB, which produces more full-length proteins than wild-type (*Figure 1—figure supplement 1*), did not increase the phenotypic penetrance (*Figure 4H*). We note that the quantification of glomerular mistargeting, as a binary binning, is less sensitive than the fluorescence measurement in quantifying trajectory choice. Thus, the resolution of this assay may not be sufficient to detect any small effect here.

Collectively, PlexB-mediated synaptic partner selection engages both extracellular and cytoplasmic modules for signaling, resembling trajectory choice but not axon fasciculation. Moreover, extracellular cleavage of PlexB does not appear to be critical in synaptic partner selection.

## Discussion

Our systematic in vivo analysis shows the divergent engagement of different PlexB structural motifs in distinct neurodevelopmental processes (*Figure 5*), arguing against a singular signaling mechanism for PlexB. We further identify cleavage as a regulatory mechanism of PlexB signaling in vivo, highlighting the functional significance of the evolutionarily conserved cleavage of class B Plexins. These experiments reveal how a single molecule, PlexB, achieves functional versatility in neurodevelopment by diversified and task-specific motif engagement, in conjunction with temporally-regulated expression and level-dependent signaling as we previously discovered (*Li et al., 2018b*).

### Differential engagement of structural motifs in distinct developmental tasks

The extracellular Sema domain is highly conserved in all Plexins and Semaphorins (*Goodman et al., 1999*). Structural and biochemical studies have highlighted its central role in mediating Plexin-Semaphorin interactions (*Janssen et al., 2010*; *Nogi et al., 2010*). Consistently, we found that all three

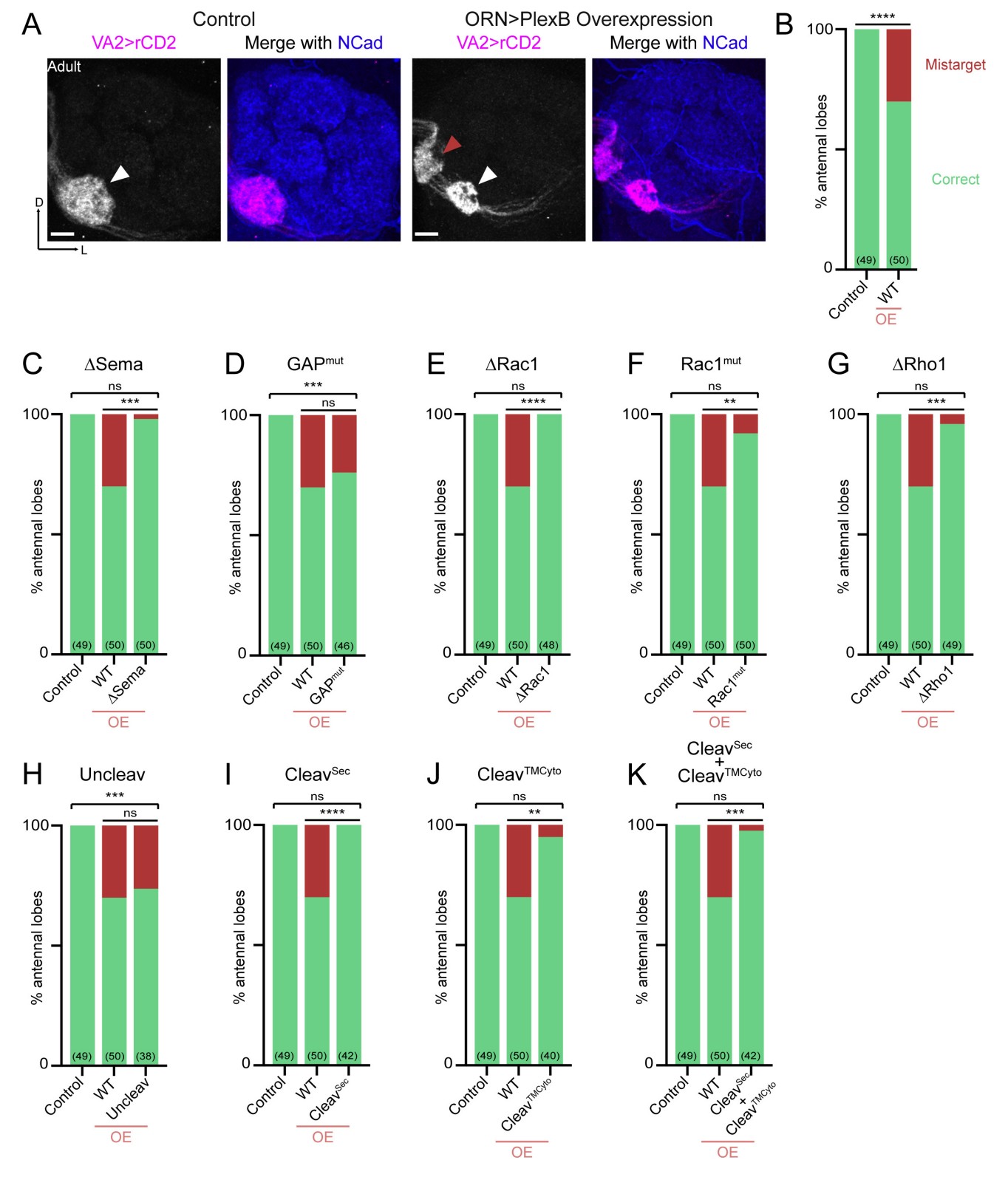

**Figure 4.** Synaptic partner selection engages both extracellular and cytoplasmic motifs of PlexB. (**A**) In *wild-type* fly brains, Or92a+ ORN axons exclusively innervate the VA2 glomerulus at the ventromedial corner of an antennal lobe (left panels; white arrowhead). Overexpression of PlexB in ORNs causes stereotypical mistargeting to the medial DM5 glomerulus (right panels; red arrowhead). Or92a+ ORN axons were labeled by membrane-localized rCD2 driven by an Or92a promoter. Antennal lobes were co-stained with a neuropil marker N-cadherin (NCad). (**B**) Penetrance of glomerular

*Figure 4 continued on next page*

*Figure 4 continued*

mistargeting in *wild-type* and PlexB overexpression brains. 'OE' hereafter denotes ORN-specific overexpression of PlexB variants. (**C–K**) Penetrance of glomerular mistargeting for respective PlexB variants. Sample sizes are noted in parentheses. Significance of the contingency tables in *Figure 4B–K* was determined by Fisher's exact test. ns, not significant; **p<0.01; ***p<0.001; ****p<0.0001. Images are shown as maximum z-projections of confocal stacks. Scale bars, 10 µm. Axes, D (dorsal), L (lateral).

DOI: https://doi.org/10.7554/eLife.48594.007

wiring steps examined here rely on the integrity of the Sema domain, further emphasizing its functional necessity.

On the other hand, the catalytic site of the RapGAP domain appears to be dispensable for all PlexB-mediated wiring processes examined. Although structural and in vitro studies have pinned

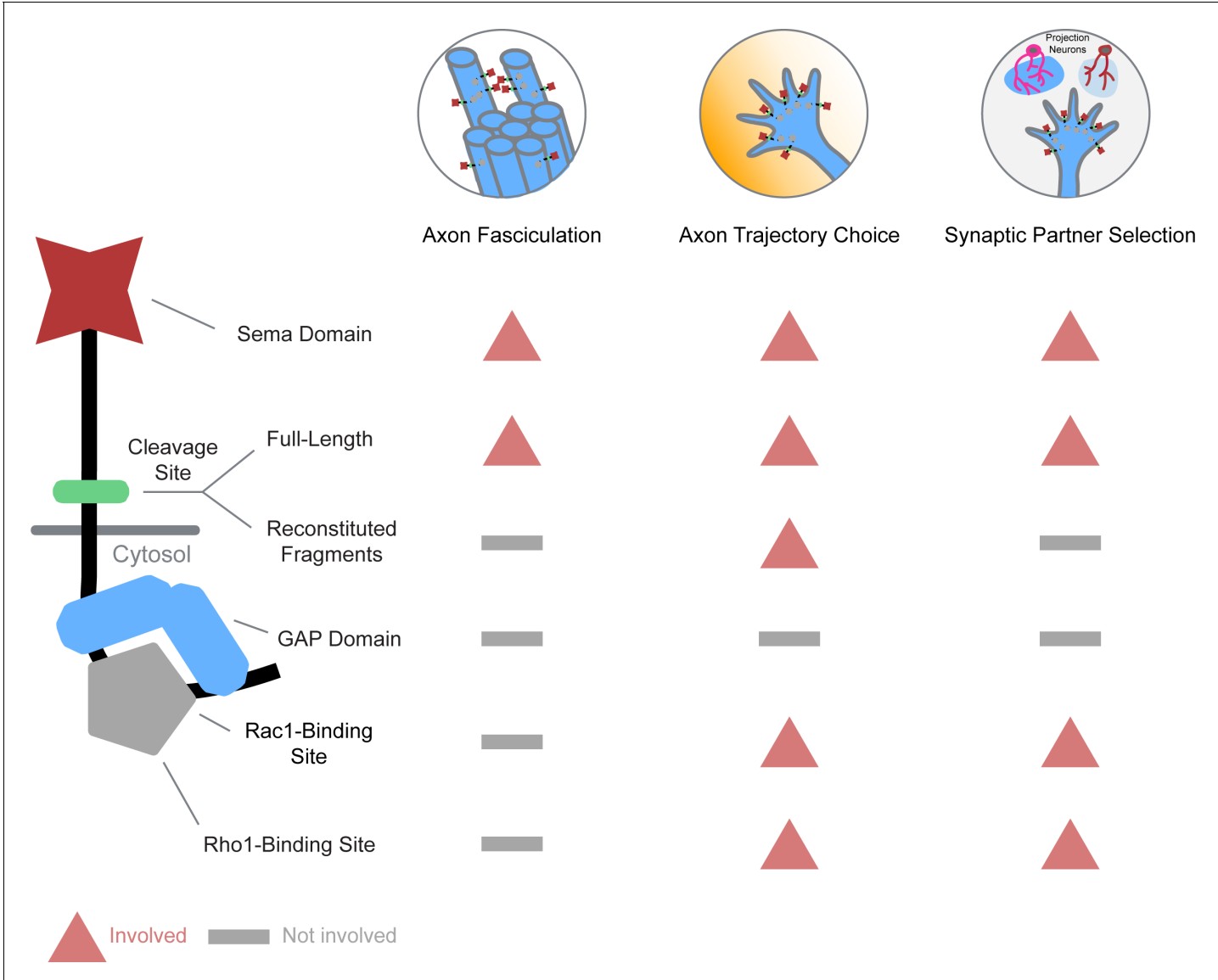

**Figure 5.** Differential engagement of PlexB structural motifs in distinct neurodevelopmental tasks. As illustrated in columns, each distinct wiring step in the development of the fly olfactory map employs a unique combination of signaling motifs. From the perspective of individual structural motifs shown in rows, each one exhibits differing importance at different developmental stages, except the universally required Sema domain and the generally expendable GAP catalytic unit.

DOI: https://doi.org/10.7554/eLife.48594.008

down Rap as the substrate for Plexin's GAP domains (*Wang et al., 2012*; *Wang et al., 2013*), in vivo studies in different developmental systems have yielded contrasting observations regarding the functional significance of its catalytic activity (*Worzfeld et al., 2014*; *Yang et al., 2016*). Thus, Rap-GAP-mediated catalysis provides one, but not the only, signaling output of Plexins, further supporting the notion that Plexins diversely engage signaling motifs for distinct developmental tasks. Although the catalytic arginine residues are conserved and are individually indispensable across species and across Plexin sub-families (*Oinuma et al., 2004*; *Rohm et al., 2000*; *Saito et al., 2009*; *Sakurai et al., 2010*; *Wang et al., 2012*; *Wang et al., 2013*), our results presented here do not exclude the possibility that *Drosophila* PlexB retains some GAP activity in our triple arginine mutant through an alternative mechanism.

It has been shown that the integrity of GTPase-binding sites is crucial for PlexB-mediated axon guidance of embryonic motor neurons (*Hu et al., 2001*). Here, we analyzed the functional involvement of these motifs in three distinct neurodevelopmental tasks and observed differential necessity among them. While the GTPase-binding sites are entirely dispensable for bundling ORN axons, both axon trajectory choice and synaptic partner selection require proper GTPase binding. Notably, the two distinct Rac1-binding sites are of different importance in mediating the PlexB-Rac1 interaction.

We note that the expression levels of our transgenes, while comparable to each other (*Figure 1—figure supplement 1*), may be different from endogenous PlexB. Thus, a negative observation (e.g., the expendability of the RapGAP catalytic unit) can possibly be caused by overexpression-induced compensation of a partial loss-of-function mutant. Editing the endogenous *PlexB* locus would overcome this caveat, at the expense of losing cell type specificity in genetic manipulation, which is of more concern for a widely expressed protein like PlexB. Despite the technical limitations, the comparative analysis here allowed us to functionally characterize PlexB structural motifs individually in vivo and reveals how the task-specific, combinatorial engagement of structural motifs enables a single molecule like PlexB to accomplish multiple distinct developmental tasks in neural circuit assembly (*Figure 5*).

## Cleavage of class B plexins – one protein, two functional forms

Numerous cell-surface proteins, including wiring molecules, are cleaved by extracellular convertases (*Duckert et al., 2004*). However, for most of them, it remains unclear how cleavage affects their signaling and functional output. For instance, all class B Plexins possess an extracellular convertase site (*Artigiani et al., 2003*), whose biological function, until now, had remained unknown. Intriguingly, cleaved PlexB fragments are not degraded but remain biochemically associated in a complex (*Artigiani et al., 2003*), suggesting that cleaved fragments participate in PlexB signaling.

We previously found that only a small fraction of endogenous PlexB proteins in brains are in a full-length form (*Li et al., 2018b*), suggesting that cleavage may play a role in tuning PlexB signaling. In the current study, we found that uncleavable PlexB produces more full-length proteins in vivo (*Figure 1—figure supplement 1*) and possesses higher activity in axon fasciculation and trajectory choice than wild-type PlexB. On the other hand, the two cleaved fragments when expressed together via separate transgenes could functionally reconstitute PlexB activity in the trajectory choice assay. Taken together, our findings support the existence of two functional forms of PlexB proteins: full-length and reconstituted fragments. Considering their distinct biophysical states, we anticipate that these two forms have different signaling properties. However, the complex and indirect readouts of in vivo developmental consequences make it difficult to quantitatively determine these properties.

Consistent with previous observations in vitro (*Artigiani et al., 2003*), we found that disrupting Rac1 or Rho1 binding reduced the presence of cleaved fragments in brains (*Figure 1—figure supplement 1*). As mutations at GTPase-binding sites did not affect the rescue of fasciculation defects (*Figure 2*), it is unlikely that these mutations disturb membrane localization, leading to insufficient cleavage. Rather, it is more likely that losing GTPase binding destabilizes the C-terminal cleaved fragment, which contains a short, degradation-prone extracellular motif. Considering that the cleaved fragments are capable of signaling, the interaction between GTPase binding and cleavage adds another layer of complexity to PlexB signaling. Disrupting GTPase binding may thus cause secondary defects by reducing cleaved PlexB.

Taken together, cleavage brings new properties and regulatory potentials to PlexB. It demands collective efforts from structural, biochemical, and functional approaches to understand this conserved feature of class B Plexins, as well as many other cleavable wiring molecules.

# Materials and methods

**Key resources table**

| Reagent type (species) or resource | Designation | Source or reference | Identifiers | Additional information |
|---|---|---|---|---|
| Genetic reagent (*Drosophila melanogaster*) | *C155-GAL4* | *Lin and Goodman, 1994* | | |
| Genetic reagent (*D. melanogaster*) | *Pebbled-GAL4* | *Sweeney et al., 2007* | | |
| Genetic reagent (*D. melanogaster*) | *Or92a-rCD2* | *Li et al., 2018b* | | |
| Genetic reagent (*D. melanogaster*) | *UAS-mtdTomato* | *Potter et al., 2010* | RRID:BDSC_30124 | |
| Genetic reagent (*D. melanogaster*) | *plexB$^{KG00878}$* | *Bellen et al., 2004* | RRID:BDSC_14579 | |
| Genetic reagent (*D. melanogaster*) | *UAS-PlexB (WT)* | this study | | Transgenic flies of UAS-PlexB variants, described in *Figure 1B* |
| Genetic reagent (*D. melanogaster*) | *UAS-PlexB (ΔSema)* | this study | | |
| Genetic reagent (*D. melanogaster*) | *UAS-PlexB (GAP$^{mut}$)* | this study | | |
| Genetic reagent (*D. melanogaster*) | *UAS-PlexB (ΔRac1)* | this study | | |
| Genetic reagent (*D. melanogaster*) | *UAS-PlexB (Rac1$^{mut}$)* | this study | | |
| Genetic reagent (*D. melanogaster*) | *UAS-PlexB (ΔRho1)* | this study | | |
| Genetic reagent (*D. melanogaster*) | *UAS-PlexB (Uncleav)* | this study | | |
| Genetic reagent (*D. melanogaster*) | *UAS-PlexB (Cleav$^{Sec}$)* | this study | | |
| Genetic reagent (*D. melanogaster*) | *UAS-PlexB (Cleav$^{TMCyto}$)* | this study | | |
| Antibody | rat anti-Ncad | Developmental Studies Hybridoma Bank | RRID:AB_528121 | 1:40 in 5% normal donkey serum |
| Antibody | rabbit anti-DsRed | Clontech | RRID:AB_10013483 | 1:200 in 5% normal donkey serum |
| Antibody | mouse anti-rat CD2 | Bio-Rad | RRID:AB_321238 | 1:200 in 5% normal donkey serum |
| Antibody | mouse anti-V5 | Thermo Fisher | RRID:AB_2556564 | |

*Continued on next page*

*Continued*

| Reagent type (species) or resource | Designation | Source or reference | Identifiers | Additional information |
|---|---|---|---|---|
| Software | ZEN | Carl Zeiss | RRID:SCR_013672 | |
| Software | ImageJ | National Institutes of Health | RRID:SCR_003070 | |
| Software | Prism | GraphPad | RRID:SCR_002798 | |
| Software | Photoshop | Adobe | RRID:SCR_014199 | |
| Software | Illustrator | Adobe | RRID:SCR_010279 | |

## *Drosophila* stocks and genotypes

Flies were raised on standard cornmeal medium with a 12 hours/12 hours light cycle at 25°C (excepting experiments described in *Figure 4*, where 29°C was used for enhanced transgenic expression). The following lines were used: *C155-GAL4* (pan-neuronal) (*Lin and Goodman, 1994*), *Pebbled-GAL4* (*Peb-GAL4*, pan-ORN) (*Sweeney et al., 2007*), *Or92a-rCD2* (VA2 ORNs) (*Li et al., 2018b*), *UAS-mtdTomato* (*Potter et al., 2010*), *plexB*$^{KG00878}$ (PlexB mutant) (*Bellen et al., 2004*), as well as our newly generated UAS transgenes encoding PlexB variants: WT, ΔSema, GAP$^{mut}$, ΔRac1, Rac1$^{mut}$, ΔRho1, Uncleav, Cleav$^{Sec}$, and Cleav$^{TMCyto}$. Complete genotypes of figure panels are described in *Supplementary file 1*.

## Generation of UAS transgenes encoding PlexB variants

The sequence encoding wild-type PlexB (*Joo et al., 2013*) was amplified by Q5 hot-start high-fidelity DNA polymerase (New England Biolabs, Ipswich, MA, USA) and assembled into a *pUAST-attB* vector (*Li et al., 2017*) by NEBuilder HiFi DNA assembly master mix (New England Biolabs, Ipswich, MA, USA). A V5 tag was inserted before the stop codon by Q5 site-directed mutagenesis kit (New England Biolabs, Ipswich, MA, USA). Afterwards, deletions and point mutations were introduced by Q5 site-directed mutagenesis kit (New England Biolabs, Ipswich, MA, USA). All constructs were transformed into NEB stable competent *E. coli* (New England Biolabs, Ipswich, MA, USA), extracted by QIAprep spin miniprep kit (QIAGEN, Hilden, Germany), and verified by full-length sequencing (Elim Biopharmaceuticals, Hayward, CA, USA). Constructs were then injected into *vas-int.Dm;;ZH-attP-86Fb* embryos (*Bischof et al., 2007*). *White+* progenies were individually balanced by *TM3* or *TM6B*, with the *vas-int.Dm* transgene removed.

## Immunocytochemistry

Fly brains were dissected and immunostained according to previously described methods (*Wu and Luo, 2006*; *Wu et al., 2017*). Briefly, brains were dissected in phosphate buffered saline (PBS) (Thermo Fisher, Waltham, MA) and subsequently fixed in 4% paraformaldehyde (Electron Microscopy Scineces, Hatfield, PA, USA) in PBS with 0.015% Triton X-100 (Sigma-Aldrich, St. Louis, MO, USA) for 20 min on a nutator at room temperature. Once fixed, brains were washed with PBST (0.3% Triton X-100 in PBS) four times, each time for 20 min on a nutator at room temperature. Brains were then blocked in 5% normal donkey serum (Jackson ImmunoResearch, West Grove, PA, USA) in PBST overnight at 4°C or for 1 hour at room temperature on a nutator. Then, brains were incubated in primary antibody diluted in the blocking solution for 36–48 hours on a 4° C nutator. Brains were then washed 4 times in PBST, each time nutating for 20 min at room temperature. Next, brains were incubated with secondary antibodies diluted in the blocking solution and nutated in the dark for 36–48 hours at 4°C. Brains were again washed with PBST four times, each time on a nutator for 20 min at room temperature. Once immunostained, brains were mounted on slides with SlowFade antifade reagent (Thermo Fisher, Waltham, MA, USA) and stored at 4° C prior to imaging.

Primary antibodies used in this study include: rat anti-NCad (1:40; DN-Ex#8, Developmental Studies Hybridoma Bank, Iowa City, IA, USA), rabbit anti-DsRed (1:200; 632496, Clontech, Mountain View, CA, USA), mouse anti-rat CD2 (1:200; OX-34, Bio-Rad, Hercules, CA, USA). Donkey secondary antibodies conjugated to Alexa Fluor 405/568/647 (Jackson ImmunoResearch, West Grove, PA, USA) were used at 1:250.

## Image acquisition, processing, and quantification

Images were acquired by a Zeiss LSM 780 laser-scanning confocal microscope (Carl Zeiss, Oberkochen, Germany), with a 40x/1.4 Plan-Apochromat oil objective (Carl Zeiss, Oberkochen, Germany). Confocal z-stacks were obtained by 1 μm intervals at the resolution of 512 × 512.

For quantification of fasciculation defects, a single scorer binned antennal lobes into three categories – 'no defasciculation', 'mild defasciculation', and 'severe defasciculation' – while blinded to the genotypes. Antennal lobes with clear trajectories and lacking axon invasion into the lobe were binned as 'no defasciculation'. Both 'mild defasciculation' and 'severe defasciculation' indicate axon invasion into the antennal lobe, while the 'severe' cases also showed the loss of trajectories.

We quantified ORN axon trajectories at 24hAPF as previously described (*Li et al., 2018b*). Briefly, the z-stack of an antennal lobe was collapsed to one image by maximum intensity projection (ZEN software, Carl Zeiss, Oberkochen, Germany). Each antennal lobe was divided into two halves (DL and VM) by the line from the ORN axon entry point to the commissure merging point. The fluorescence intensities of the DL and VM halves and an area outside of the antennal lobe (background) were measured by ImageJ (NIH, Bethesda, MD, USA). Background fluorescence intensity was deducted to obtain the corrected intensities of the DL and VM axon trajectories. The DL/VM ratio was calculated by Excel (Microsoft, Redmond, WA, USA).

Images were exported as maximum projections by ZEN (Carl Zeiss, Oberkochen, Germany) in the format of TIFF. Preview (Apple, Cupertino, CA, USA) was used for image rotation and cropping. Illustrator (Adobe, San Jose, CA) was used to make diagrams and assemble figures.

## Western blot

Brains and ventral nerve cords of third-instar larvae were dissected in the Schneider's *Drosophila* medium (Thermo Fisher, Waltham, MA, USA) and snap frozen in liquid nitrogen before stored at – 80°C. Samples were lysed on ice in pre-cooled RIPA buffer (Thermo Fisher, Waltham, MA, USA) with protease inhibitors (100X Halt cocktail; Thermo Fisher, Waltham, MA, USA) and then rotated for 2 hours at 4°C. After centrifugation for 30 min at 16000 RCF (relative centrifugal force) at 4°C, the supernatant was collected and kept on ice. Laemmli sample buffer (Bio-Rad, Hercules, CA, USA) and 20 mM dithiothreitol (Sigma-Aldrich, St. Louis, MO, USA) were added to the sample, followed by heating at 95°C for 10 min. Precision Plus Protein Kaleidoscope prestained protein standard (Bio-Rad, Hercules, CA, USA) was used as the molecular weight marker. Electrophoresis with the NuPAGE Tris-acetate gel and PVDF membrane transfer (Thermo Fisher, Waltham, MA, USA) were performed according to the manufacturer's protocols. We note that the PlexB protein level is extremely low in vivo, even in the context of overexpression (*Li et al., 2018b*). Accordingly, routine blocking reagents, such as nonfat dry milk or bovine serum albumin, and conventional substrates for HRP were not able to yield clear blotting results. The membrane was blocked by TBS-buffered SuperBlock solution (Thermo Fisher, Waltham, MA, USA) and incubated with the primary antibody (mouse anti-V5, 1:300, R960-25; Thermo Fisher, Waltham, MA, USA) in SuperBlock for 72 hours on a 4°C orbital shaker. After washing with TBST (25 mM Tris, 0.15M NaCl, 0.05% Tween-20, pH 7.5; Thermo Fisher, Waltham, MA, USA), the membrane was incubated with the secondary antibody (goat anti-mouse HRP-conjugated, 1:2500; Thermo Fisher, Waltham, MA, USA) for 2 hours on an orbital shaker at room temperature. The signal was developed with Clarity Max Western ECL substrate (Bio-Rad, Hercules, CA, USA) and captured by the ChemiDoc XRS+ system (Bio-Rad, Hercules, CA, USA). Afterwards, the membrane was stripped in Restore PLUS Western blot stripping buffer for 15 min at 37°C with occasional shaking, followed by re-blocking with TBS-buffered SuperBlock. N-cadherin and actin controls were blotted in a routine Western procedure with the following antibodies: rat anti-NCad (1:300; DN-Ex#8, Developmental Studies Hybridoma Bank, Iowa City, IA, USA) and mouse anti-actin (1:2000; ab8224, Abcam, Cambridge, UK).

## Statistical analysis

No statistical methods were used to determine sample sizes, but our sample sizes were similar to those generally employed in the field. Antennal lobes damaged in dissection were excluded from analysis; otherwise, all samples were included. Except for scoring the fasciculation defects, data collection and analysis were not performed blind to the conditions of the experiments. GraphPad Prism (GraphPad Software, La Jolla, CA, USA) was used for statistical analysis and plotting. Significance

among multiple groups was determined by one-way ANOVA with Tukey's test for multiple comparisons. Significance of contingency tables was determined by Fisher's exact test.

## Acknowledgements

We thank Alex Kolodkin for the kind gifts of reagents, the Bloomington *Drosophila* Stock Center and the Vienna *Drosophila* Resource Center for fly lines, and Addgene for plasmids. We acknowledge Thomas Clandinin, K Christopher Garcia, Yanyang Ge, Justus Kebschull, Hongjie Li, Tongchao Li, Colleen McLaughlin, Daniel Pederick, Kang Shen, Andrew Shuster, Michael Simon, Yukari Takeo, David Wang, Qijing Xie, Chuanyun Xu, and all Luo lab members for technical support, insightful advice, and/or critical comments on this study. We thank Alice Ting for her support of Shuo Han. We also acknowledge Mary Molacavage and Stephanie Wheaton for administrative assistance.

## Additional information

### Funding

| Funder | Grant reference number | Author |
|---|---|---|
| Howard Hughes Medical Institute | | Liqun Luo |
| National Institutes of Health | R01-DC005982 | Liqun Luo |
| The Yingwei Cui and Hui Zhao Neuroscience Fund | | Liqun Luo |
| Stanford University | Bio-X Undergraduate Summer Research Program | Ricardo Guajardo |
| Stanford University | Stanford Undergraduate Advising and Research Major Grant | Ricardo Guajardo |
| Stanford University | Bio-X Bowes Interdisciplinary Graduate Fellowship | Shuo Han |
| Genentech Foundation | Genentech Foundation Predoctoral Fellowship | Jiefu Li |
| Stanford University | Vanessa Kong-Kerzner Graduate Fellowship | Jiefu Li |

The funders had no role in study design, data collection and interpretation, or the decision to submit the work for publication.

### Author contributions

Ricardo Guajardo, Resources, Data curation, Formal analysis, Validation, Investigation, Visualization, Methodology, Writing—original draft, Writing—review and editing; David J Luginbuhl, Resources, Investigation; Shuo Han, Investigation; Liqun Luo, Conceptualization, Resources, Supervision, Funding acquisition, Methodology, Writing—original draft, Project administration, Writing—review and editing; Jiefu Li, Conceptualization, Resources, Data curation, Formal analysis, Supervision, Validation, Investigation, Visualization, Methodology, Writing—original draft, Writing—review and editing

### Author ORCIDs

Ricardo Guajardo (iD) https://orcid.org/0000-0003-2379-3480
Liqun Luo (iD) https://orcid.org/0000-0001-5467-9264
Jiefu Li (iD) https://orcid.org/0000-0002-0062-4652

### Decision letter and Author response

Decision letter https://doi.org/10.7554/eLife.48594.013
Author response https://doi.org/10.7554/eLife.48594.014

## Additional files

### Supplementary files

• Supplementary file 1. Genotypes of flies in each experiment.
DOI: https://doi.org/10.7554/eLife.48594.009

• Transparent reporting form
DOI: https://doi.org/10.7554/eLife.48594.010

### Data availability

All data generated or analysed during this study are included in the manuscript and supporting files.

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
