## [Decision Letter]

Thank you for submitting your article "Functional divergence of Plexin B structural motifs in distinct steps of *Drosophila* olfactory circuit assembly" for consideration by *eLife*. Your article has been reviewed by three peer reviewers, and the evaluation has been overseen by Kristin Scott as the Reviewing Editor and K VijayRaghavan as the Senior Editor. The following individuals involved in the review of your submission have agreed to reveal their identity: Takahiro Chihara (Reviewer #1); Bing Ye (Reviewer #2).

The reviewers have discussed the reviews with one another and the Reviewing Editor has drafted this decision to help you prepare a revised submission.

Summary:

Guajardo et al. provide in vivo evidence for the functional requirements of Plexin B functional motifs in *Drosophila* olfactory receptor neurons, ORNs. The authors generated a series of transgenic lines expressing mutated Plexin B and performed genetic rescue experiments for axon bundle formation (fasciculation) and overexpression in ORNs for axon trajectory choice and synaptic partner selection. The authors found differential requirements of the Plexin B domains. For example, the Sema domain is required for all the steps but the GTPase-binding domains are involved in trajectory choice and synaptic partner choice but not in axon fasciculation. The study nicely builds on the authors' previous work to elucidate novel concepts. It also offers an in vivo test of ideas based on previous biochemical findings (e.g., the function of the RapGAP domain). The study is well designed and the results are convincing. The paper is clearly written and requires only minor text changes.

Essential revisions:

1) The authors find that the GAP domain is not involved in any steps of ORN assembly. Is there experimental evidence showing GAP activity is really abolished in the *Drosophila* Plexin B mutant GAP^mut^ (R1570A, R1571G, R1899A)? The authors should indicate the relevant references or experimental evidence or consider the possibility that activity is not abolished in the text.

2) The findings and discussion regarding the cleavage of Plexin B are intriguing. Based on the results, the uncleavable Plexin B is more effective (functional) than wild-type Plexin B (except glomerular selection). The authors discussed the potential physiological significance of Plexin B cleavage, stating that "Plexin B cleavage is a way of tuning the Plexin B level (function)". Can authors provide any genetic evidence or references supporting this claim? More discussion here is warranted.

---

## [Author Response]

Essential revisions:1) The authors find that the GAP domain is not involved in any steps of ORN assembly. Is there experimental evidence showing GAP activity is really abolished in the Drosophila Plexin B mutant GAP^mut^ (R1570A, R1571G, R1899A)? The authors should indicate the relevant references or experimental evidence or consider the possibility that activity is not abolished in the text.

We agree with the reviewers that we do not have direct evidence on the GAP catalytic activity of *Drosophila* Plexin B and its loss in this mutant. These three arginine residues are highly conserved across species and across Plexin sub-families (see the phylogenetic alignment, which we added as a new Figure 1—figure supplement 2). Previous studies (Wang et al., 2012, 2013) assayed Plexins from different species and sub-families and found that these residues, individually, are indispensable for the GAP catalytic activity. Thus, the combined triple mutant (R1570A, R1571G, R1899A) is likely to disrupt the GAP activity. However, we cannot exclude the possibility that *Drosophila* Plexin B uses unconventional and currently unknown residues in catalysis. We have added discussion on this possibility in the revised manuscript

2) The findings and discussion regarding the cleavage of Plexin B are intriguing. Based on the results, the uncleavable Plexin B is more effective (functional) than wild-type Plexin B (except glomerular selection). The authors discussed the potential physiological significance of Plexin B cleavage, stating that "Plexin B cleavage is a way of tuning the Plexin B level (function)". Can authors provide any genetic evidence or references supporting this claim? More discussion here is warranted.

The western blot in Figure 1—figure supplement 1 shows elevated level of full-length Plexin B proteins when the uncleavable Plexin B is expressed, indicating that cleavage provides a way to down-regulate full-length Plexin B. Phenotypically, disrupting cleavage intensifies trajectory shifting caused by elevating the Plexin B level (via overexpression). We have modified the text to make these conclusions clearer.